# Depression, anxiety, stress and their associated factors among Ethiopian University students during an early stage of COVID-19 pandemic: An online-based cross-sectional survey

Wudneh Simegn[1], Baye Dagnew[2], Yigizie Yeshaw[2,3], Sewbesew Yitayih[4], Birhanemeskel Woldegerima[1], Henok Dagne[5]*

1 Department of Pharmaceutics and Social Pharmacy, School of Pharmacy, College of Medicine and Health Sciences, University of Gondar, Gondar, Ethiopia, 2 Department of Human Physiology, School of Medicine, College of Medicine and Health Sciences, University of Gondar, Gondar, Ethiopia, 3 Department of Epidemiology and Biostatistics, Institute of Public Health, College of Medicine and Health Sciences, University of Gondar, Gondar, Ethiopia, 4 Department of Psychiatry, School of Medicine, College of Medicine and Health Sciences, University of Gondar, Gondar, Ethiopia, 5 Department of Environmental and Occupational Health and Safety, Institute of Public Health, College of Medicine and Health Sciences, University of Gondar, Gondar, Ethiopia

* enoch2313@gmail.com

**Data Availability Statement:** The data underlying the results presented in the study are available and will be shared during publication.

## Abstract

### Background

The occurrence of the Coronavirus Disease 2019 (COVID-19) affects the mental health situation of almost everyone, including University students who spent most of their time at home due to the closure of the Universities. Therefore, this study aimed at assessing depression, anxiety, stress and identifying their associated factors among university students in Ethiopia during the early stage of the COVID-19 pandemic.

### Methods

We invited students to complete an online survey using Google forms comprising consent, socio-demographic characteristics, and the standard validated depression, anxiety, and stress scale (DASS-21) questionnaire. After completion of the survey from June 30 to July 30, 2020, we exported the data into SPSS 22. Both descriptive and analytical statistics were computed. Associated factors were identified using binary logistic regression and variables with a p-value <0.05 were declared as statistically significant factors with the outcome variables.

### Results

A total of 423 students completed the online survey. The prevalence of depression, anxiety, and stress in this study was 46.3%, 52%, and 28.6%, respectively. In the multivariable model, female sex, poor self-efficacy to prevent COVID-19, those who do not read any

**Funding:** The authors received no specific funding for this work.

**Competing interests:** The authors have declared that no competing interests exist.

**Abbreviations:** AOR, Adjusted Odds Ratio; COR, Crude Odds Ratio; COVID-19, Coronavirus Disease; DASS, Depression Anxiety Stress Scale; SPSS, Statistical Package for Social Sciences.

material about COVID-19 prevention, lack of access to reading materials about their profession, and lack of access to uninterrupted internet access were significantly associated with depression. Female sex, lower ages, students with non-health-related departments, those who do not think that COVID-19 is preventable, and those who do not read any materials about COVID-19 prevention were significantly associated with anxiety. Whereas, being female, students attending 1st and 2nd years, those who do not think that COVID-19 is preventable, presence of confirmed COVID-19 patient at the town they are living in, and lack of access to reading materials about their profession were significantly associated with stress.

## Conclusions

Depression, anxiety, and stress level among University students calls for addressing these problems by controlling the modifiable factors identified and promoting psychological well-being of students.

## Background

The Coronavirus Disease 2019 (COVID-19) is a viral pandemic that emerged for the first time in Wuhan, China, and spreads all over the world between December 2019 and early 2020 [1]. The virus has resulted in more than 54 million cases and 1.3 million deaths worldwide [2].

Because of the sudden nature of the outbreak and the infectious power of the virus, it will inevitably cause serious threats to people's physical health and lives. It has also triggered a wide variety of psychological problems, such as panic disorder, anxiety, depression, and stress [1, 3, 4]. Depression, anxiety, and stress affect the outcome of chronic diseases such as diabetes mellitus, cardiovascular diseases, cancer, and obesity [5]. Depression, anxiety and stress can affect every population including students all of which can affect job performance, quality of sleep, routine activities, and productivity of the victims [6].

The prevalence of depression, anxiety and stress is high among the general population during the COVID-19 pandemic [7]. This number is expected to be higher among University students during the COVID-19 pandemic as they are exposed to excessive working hours, living in a competitive academic environment, and financial problems [8]. According to a study in Canada, the prevalence of depression, anxiety and stress is 39.5%, 23.8% and 80.3%, respectively [9]. The prevalence of depression, anxiety and stress among European College students was 39.0%, 47.0%, and 35.8%, respectively [10]. In Saudi Arabia, 58.1% of the University's academic community had anxiety and 50.2% of them had depression [11]. In Pakistan, 57.6% of Medical students had depression, 74% anxiety and 57.7% [12]. Similarly, another study in Pakistan revealed that 48%, 68.54% and 53.2% of students had depression, anxiety, and stress, respectively [13]. The prevalence of depression, anxiety, and stress among Malaysian undergraduate students was 30.7%, 55.5%, and 16.6%, respectively [14]. In Ethiopia, according to a study in Addis Ababa medical students, 51.30% of them had depression and 30.10% had anxiety symptoms [15].

Previous studies revealed that internet access [16–18], self-efficacy [19–21], self-rated health [9], age, marital status, and sex of the students were significantly associated with depression [22]. Similarly, age [15, 22–32], year of study, and social support [15], marital status, and sex of the students were significantly associated with anxiety [15, 22]. Academic performance [9], age, marital status [22], and sex [1, 23, 29, 33–42] were determinants of stress. Though

COVID-19 may take a significant human toll as well as causes public fear, economic loss, and other adverse outcomes as mentioned earlier, it is common for health professionals and managers to focus predominantly on disease prevention and treatment, leaving/neglecting the psychological and psychiatric implications secondary to the phenomenon. This leads to a gap in coping strategies and increases the burden of associated diseases [43].

Therefore, understanding and investigating the public psychological states during this tumultuous time is of practical significance. Hence, this study aimed to determine the magnitudes of depression, anxiety, stress and their associated factors among University Students in Ethiopia. This study will provide a concrete basis for tailoring and implementing relevant mental health intervention policies to cope with the challenge of the outbreak efficiently and effectively.

## Methods

### Study area, design and period

This online cross-sectional survey was conducted among university students in Ethiopia. The actual data collection period was from June 30 to July 30, 2020.

### Population and eligibility criteria

We included all University students who were using social media such as Facebook, Twitter, Instagram, and who were voluntary to fill out the survey form. We preferred to use social media users because it enables us to collect the data without direct contact with the study participants, which is crucial to reduce the rate of spread of the COVID 19 pandemic. The flow chart of study participants is included below (**Fig 1**).

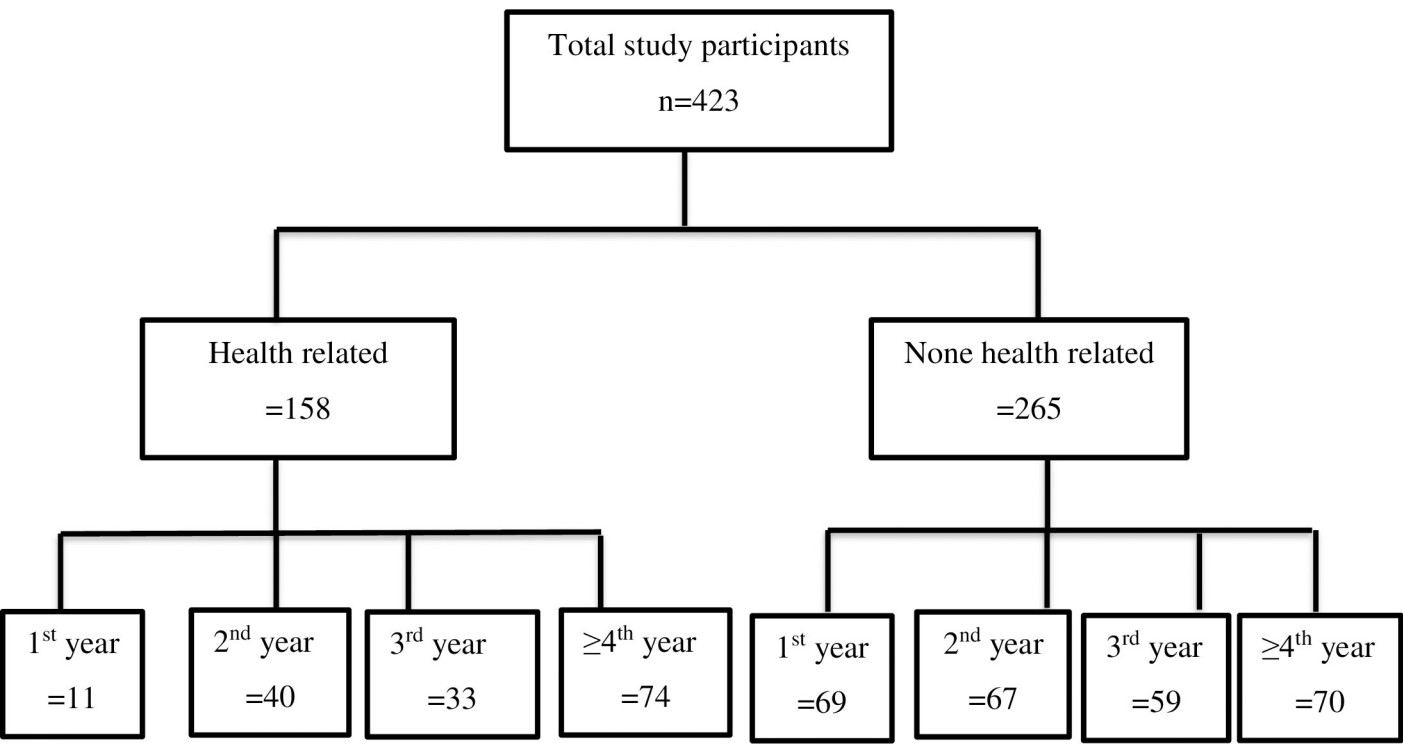

**Fig 1. Flowchart of study participants.**

### Sample size determination and sampling technique

The sample size was determined using the single population proportion formula by considering the following assumptions; proportion (p) = 0.5 (since there was no previous study), 95% confidence level, the margin of error of 5%, and 10% non-response rate. The minimum sample size was 384 and after adding a non-response rate, we found a final sample size of 423.

### Data collection questionnaire and procedure

We used Lovibond's short version of the DASS-21 (Depression, Anxiety, and Stress Scale-21) which is a psychological screening instrument capable of differentiating symptoms of depression, anxiety, and stress [44]. Each of the three DASS-21 scales contains 7 items, divided into subscales with similar content. The cut-off values are explained elsewhere [44]. A single item was used to assess the levels of self-efficacy related to COVID-19: "How confident are you that you can prevent getting COVID-19 in case of an outbreak"[45].

### Data management and statistical analysis

Data were collected using Google Forms and SPSS version 22 was used for statistical analysis. Categorical variables were expressed in terms of frequency and percent whereas continuous variables were described by the mean and standard deviation. Binary logistic regression was employed to identify associated factors of depression, anxiety and, stress. First, we performed bivariable binary logistic regression to identify candidate variables for the final analysis using p-value <0.2 as a cut-off point. Then, multivariable logistic regression was carried out to decide statistically significant variables of depression, anxiety, and stress at p-value<0.05.

### Ethical approval and consent to participate

Ethical approval was obtained from the University of Gondar Ethical Review Board. Consent was obtained from each study participant to assure their willingness to participate online and no identifiers were listed in the questionnaire to make it confidential. This study was conducted according to the declaration of Helsinki.

## Results

### Socio-demographic characteristics

Four hundred and twenty-three students participated in the study with a response rate of 100%. Two hundred and seventy-two (64.3%) were males and the mean age of study participants was 22.96 years (range: 18–34). One hundred and fifty-eight (37.4%) participants attended health-related departments and 34.0% of students were 4th year and above followed by the second year (25.3) and third-year (21.7%). Two hundred and twenty-nine (54.1%) participants reported that they had good-self efficacy. Two hundred and twenty-one (52.2%) participants thought that COVID-19 was preventable and 241 (57.0%) participants reported having a clear information source about COVID-19. Two hundred and eighty-five (67.4%) participants reported the presence of confirmed COVID-19 patient in the town they were living. Two hundred and ninety-one (68.8%) participants had reading materials about their profession and the majority of the participants (84.2%) had internet access (**Table 1**).

**Table 1. Socio-demographic characteristics of the study participants of depression, anxiety, stress among Ethiopian University students during an early stage of COVID-19, 2020 (N = 423).**

| Variables | Categories | Frequency | Percent |
|---|---|---|---|
| Sex | Female | 151 | 35.7 |
| | Male | 272 | 64.3 |
| Age in years | 18–21 | 111 | 26.2 |
| | 22–23 | 141 | 33.3 |
| | 24–25 | 96 | 22.7 |
| | 26 and above | 75 | 17.7 |
| Department | Health related | 158 | 37.4 |
| | Other than health | 265 | 62.6 |
| Year of study | 1st | 80 | 19.0 |
| | 2nd | 107 | 25.3 |
| | 3rd | 92 | 21.7 |
| | 4th and above | 144 | 34.0 |
| How do you rate yourself to protect yourself from COVID 19? | Not prepared | 194 | 45.9 |
| | Prepared | 229 | 54.1 |
| Do you think COVID-19 is preventable? | Yes | 221 | 52.2 |
| | No | 202 | 47.8 |
| Is there a clear information source that you can easily access about COVID 19? | Yes | 241 | 57.0 |
| | No | 182 | 43.0 |
| Do you feel that you are well protected from COVID-19 in your living area? | Yes | 174 | 41.1 |
| | No | 249 | 58.9 |
| Have you ever read any materials regarding the prevention of COVID-19? | Yes | 250 | 59.1 |
| | No | 173 | 40.9 |
| Is there any confirmed COVID-19 patient in the town you are living in? | Yes | 285 | 67.4 |
| | No | 138 | 32.6 |
| Have you got any reading materials about your profession? | Yes | 291 | 68.8 |
| | No | 132 | 31.2 |
| Can you access uninterrupted internet service? | Yes | 356 | 84.2 |
| | No | 67 | 15.8 |

## Prevalence of depression, anxiety, and stress

One hundred and ninety–six (46.3% (95% CI: 41.6%, 50.8%), 220 (52% (95% CI: 47.1%, 56.7%)) and 121 (28.6% (24.6%, 32.9%)) participants had depression, anxiety, and stress, respectively (**Table 2**).

## Factors associated with depression

Sex, age, department, self-efficacy, perception of whether COVID-19 is preventable, presence of easily accessible information source about COVID-19 prevention, self-rated preparation prevent from COVID-19, ever accessed any materials regarding prevention of COVID-19, presence any confirmed COVID-19 patient at the town of living, access to any reading materials about the profession, and access to uninterrupted internet were candidate variables for multivariable logistic regression (p-value<0.2). In the final model; female sex (AOR = 2.20; 95% CI: 1.25–3.86), not prepared to protect themselves from COVID-19 (AOR = 2.87; 95% CI: 1.59–5.20), not ever read any materials regarding prevention of COVID-19 (AOR = 2.25; 95% CI: 1.21–4.1)), had not any reading materials about profession (AOR = 2.38; 95% CI: 1.23–4.59), and lack of access to internet (AOR = 3.32; 95% CI: 1.34–8.21) were significantly associated with depression (**Table 3**).

**Table 2. Prevalence of depression, anxiety and stress among Ethiopian University students in Ethiopia during an early stage of COVID-19 pandemic 2020 (N = 423).**

| | | | |
|---|---|---|---|
| Depression | Normal | 227 | 53.7 |
| | Mild | 53 | 12.5 |
| | Moderate | 72 | 17.0 |
| | Sever | 22 | 5.2 |
| | Extremely sever | 49 | 11.6 |
| | Total with depression | 196 | 46.3% (95% CI: 41.6%, 50.8%) |
| Anxiety | Normal | 203 | 48.0 |
| | Mild | 35 | 8.3 |
| | Moderate | 74 | 17.5 |
| | Sever | 28 | 6.6 |
| | Extremely sever | 83 | 19.6 |
| | Total with anxiety | 220 | 52% (95% CI: 47.1%, 56.7%) |
| Stress | Normal | 302 | 71.4 |
| | Mild | 38 | 9.0 |
| | Moderate | 29 | 6.9 |
| | Sever | 33 | 7.8 |
| | Extremely sever | 21 | 5.0 |
| | Total with stress | 121 | 28.6% (95% CI: 24.6%, 32.9%) |

**Table 3. Factors associated with depression among Ethiopian University students during the early phase of COVID-19 pandemic, 2020 (N = 423).**

| Variables | Categories | Depression | | COR (95% CI) | AOR (95% CI) |
|---|---|---|---|---|---|
| | | Yes | No | | |
| Sex | Female | 96(63.6) | 55(36.4) | 3.00(1.98,4.54) | 2.20(1.25,3.86)* |
| | Male | 100(36.8) | 172(63.2) | 1 | |
| Age in years | 18–21 | 81(73.0) | 30(27.0) | 9.95(4.97,19.91) | 2.32(0.96,5.63) |
| | 22–23 | 67(47.5) | 74(52.5) | 3.34(1.75,6.35) | 2.23(1.00,5.00) |
| | 24–25 | 32(33.3) | 64(66.7) | 1.84(0.91,3.70) | 1.92(0.82,4.50) |
| | 26 + | 16(21.3) | 59(78.7) | 1 | 1 |
| Department | Health related | 41(25.9) | 117(74.1) | 1 | |
| | Other than health | 155(58.3) | 110(41.5) | 4.02(2.61,6.19) | 1.56(0.86,4.50) |
| How do you rate to protect yourselves from COVID 19? | Not prepared | 144(74.2) | 50(25.8) | 9.80(6.27,15.32) | 2.87(1.59,5.20)** |
| | prepared | 52(22.5) | 177(77.3) | 1 | 1 |
| Do you think COVID-19 is preventable? | Yes | 56(25.3) | 165(74.7) | 1 | 1 |
| | No | 140(69.3) | 62(30.7) | 6.65(4.34,10.18) | 1.19(0.60,2.35) |
| Is there a clear information source that you can easily access about COVID 19? | Yes | 59(24.5) | 182(75.5) | 1 | 1 |
| | No | 137(75.3) | 45(24.7) | 9.39(6.01,14.68) | 1.72(0.82,3.59) |
| Do you feel that you are well protected from COVID-19 in your living area? | Yes | 35(20.1) | 139(79.9) | 1 | 1 |
| | No | 161(64.7) | 88(35.3) | 7.26(4.62,11.43) | 1.20(0.63,2.29) |
| Have you ever read any materials regarding the prevention of COVID-19? | Yes | 68(27.2) | 182(72.8) | 1 | 1 |
| | No | 128(74.0) | 45(26.0) | 7.6(4.90,11.81) | 2.25(1.22,4.13)* |
| Is there any confirmed COVID-19 patient at town you are living? | Yes | 118(41.4) | 167(58.6) | 1.84(1.22,2.77) | 0.95(0.53,1.69) |
| | No | 78(56.5) | 60(43.5) | 1 | 1 |
| Have you got any reading materials about your profession? | Yes | 93(32.0) | 198(68.0) | 1 | 1 |
| | No | 103(78.0) | 29(22.0) | 7.56(4.67,12.22) | 2.38(1.23,4.59)* |
| Can you access uninterrupted internet service? | Yes | 140(39.3) | 216(60.7) | 1 | 1 |
| | No | 11(16.4) | 56(83.6) | 7.85(3.97,15.51) | 3.32(1.34,8.21)* |

Hosmer and Lemeshow goodness-of-fit test p-value = 0.236

* p-value <0.05

**p<0.01.

## Factors associated with anxiety

Sex, age, department, self-rated protection from COVID-19, whether they thought COVID-19 is preventable, whether they feel that they are well protected from COVID-19 at their living area, whether they have ever read any materials regarding prevention of COVID-19, presence of any confirmed COVID-19 patient at the town they are living and access to reading materials about their profession were candidate variables for multivariable logistic regression (p-value<0.2). In the final model; female sex (AOR = 3.08; 95% CI: 1.6, 5.62), age 18–21 years (AOR = 4.78; 95% CI: 1.89, 12.09) and 22–23 years (AOR = 2.56; 95% CI: 1.99,10.42), being in none-health related departments (AOR = 2.67; 95% CI: 1.45,4.92), assuming that COVID-19 was not preventable (AOR = 3.50; 95% CI: 1.94,6.32), and did not read any materials regarding prevention of COVID-19 (AOR = 4.71; 95% CI: 2.56,8.67) were significantly associated with anxiety (**Table 4**).

## Factors associated with stress

Sex, department, years of study, self-rated protection from COVID-19, whether they thought COVID-19 is preventable, whether they feel that they are well protected from COVID-19 at their living area, whether they have ever read any materials regarding prevention of COVID-19, presence of any confirmed COVID-19 patient at the town they are living, access to reading materials about their profession, and access to uninterrupted internet service were candidate

**Table 4. Factors associated with anxiety among Ethiopian University students during early phase of COVID-19 pandemic, 2020 (N = 423).**

| Variables | Categories | Anxiety | | COR (95% CI) | AOR (95% CI) |
|---|---|---|---|---|---|
| | | Yes | No | | |
| Sex | Female | 106(70.2) | 45(29.8) | 3.26(2.13,4.98) | 3.08(1.69,5.62)*** |
| | Male | 114(41.9) | 158(58.1) | 1 | 1 |
| Age in years | 18–21 | 87(76.4) | 24(21.6) | 15.79(7.56,32.97) | 4.78(1.89,12.09)** |
| | 22–23 | 80(56.7) | 61(43.3) | 5.71(2.92,11.16) | 2.56(1.99,10.42)** |
| | 24–25 | 39(40.6) | 57(59.4) | 2.98(1.46,6.06) | 3.67(1.54,8.77)* |
| | 26 + | 14(18.7) | 61(81.3) | 1 | 1 |
| Department | Health related | 45(28.5) | 113(71.5) | 1 | |
| | Other than health | 175(66.0) | 90(34.0) | 4.88(3.18,7.49) | 2.67(1.45,4.92)* |
| How do you rate to protect yourselves from COVID 19? | Not prepared | 144(74.2) | 50(25.8) | 5.79(3.78,8.85) | 1.12(0.62,2.20) |
| | prepared | 76(33.2) | 153(66.8) | 1 | 1 |
| Do you think COVID-19 is preventable? | Yes | 57(25.8) | 164(74.2) | 1 | 1 |
| | No | 163(80.7) | 39(19.3) | 12.5(7.58,19.07) | 3.50(1.94,6.32)* |
| Do you feel that you are well protected from COVID-19 in your living area? | Yes | 48(27.6) | 126(72.4) | 1 | 1 |
| | No | 172(69.1) | 77(30.9) | 5.86(3.82,8.99) | 1.03(0.55,1.93) |
| Have you ever read any materials regarding the prevention of COVID-19? | Yes | 74(29.6) | 176(70.4) | 1 | 1 |
| | No | 146(84.4) | 27(15.6) | 12(4.90,11.81) | 4.71(2.56,8.67)*** |
| Is there any confirmed COVID-19 patient in the town you are living in? | Yes | 141(49.5) | 144(50.5) | 1 | 1 |
| | No | 79(57.2) | 59(42.8) | 1.36(0.90,2.05) | 1.12(0.64,2.09) |
| Have you got any reading materials about your profession? | Yes | 121(41.6) | 170(58.4) | 1 | 1 |
| | No | 170(58.4) | 33(25.0) | 4.21(2.66,6.66) | 1.57(0.80,3.07) |

Hosmer and Lemeshow goodness-of-fit test p-value = 0.773

* p-value <0.05

**p-value <0.01 and

*** p-value <0.001.

**Table 5. Factors associated with stress among Ethiopian University students during the early phase of COVID-19 pandemic, 2020 (N = 423).**

| Variables | Categories | Stress | | COR (95% CI) | AOR (95% CI) |
|---|---|---|---|---|---|
| | | Yes | No | | |
| Sex | Female | 67(44.4) | 84(55.6) | 3.22(2.07,4.99) | 2.26(1.27,4.03)* |
| | Male | 54(19.9) | 218(80.1) | 1 | 1 |
| Department | Health related | 20(12.7) | 138(87.3) | 1 | |
| | Other than health | 101(38.1) | 164(61.9) | 4.24(2.50,7.20) | 1.52(0.71,3.22) |
| Year of study | 1–2 | 93(49.7) | 94(50.3) | 7.35(4.51,11.96) | 3.62(2.03,6.47)*** |
| | ≥3 | 28(11.9) | 208(88.1) | 1 | 1 |
| How do you rate your to protect yourselves from COVID 19? | Not prepared | 91(46.9) | 103(53.1) | 5.86(3.64,9.43) | 1.03(0.51,2.08) |
| | Prepared | 30(13.1) | 199(86.9) | 1 | 1 |
| Do you think COVID-19 is preventable? | Yes | 18(8.1) | 203(91.9) | 1 | 1 |
| | No | 103(51.0) | 99(49.0) | 11.70(6.12,20.45) | 3.34(2.13,8.85)* |
| Do you feel that you are well protected from COVID-19 in your living area? | Yes | 18(10.3) | 156(89.7) | 1 | 1 |
| | No | 103(41.4) | 146(58.6) | 6.26(3.62,10.43) | 1.01(0.47,2.16) |
| Have you ever read any materials regarding the prevention of COVID-19? | Yes | 30(12.0) | 220(88.0) | 1 | 1 |
| | No | 91(52.6) | 82(47.4) | 8.5(5.90,13.51) | 2.15(1.12,3.99)* |
| Is there any confirmed COVID-19 patient in the town you are living in? | Yes | 66(23.2) | 219(76.8) | 2.14(1.22,3.77) | 1.81(1.01,3.28)* |
| | No | 55(39.9) | 83(60.1) | 1 | 1 |
| Have you got any reading materials about your profession? | Yes | 49(16.8) | 242(83.2) | 1 | 1 |
| | No | 72(54.5) | 60(45.5) | 5.96(3.67,9.22) | 2.17(1.15,4.07)* |
| Can you access uninterrupted internet service? | Yes | 85(23.9) | 271(76.1) | 1 | 1 |
| | No | 36(53.7) | 31(46.3) | 3.75(2.97,6.51) | 1.28(0.60,2.77) |

Hosmer and Lemeshow goodness-of-fit test p-value = 0.374

* p-value <0.05

**p-value <0.01 and

*** p-value <0.001.

variables of stress for multivariable logistic regression(p-value<0.2). In the final model; female sex (AOR = 2.26; 95% CI:1.27, 4.03), 1st to 2nd years of study (AOR = 3.62; 95% CI: 2.03, 6.47), those who do not think that COVID-19 is preventable (AOR = 3.34; 95% CI: 2.13, 8.85), those who never read any materials regarding prevention of COVID-19 (AOR = 2.15; 95% CI: 1.12, 3.99), presence of confirmed COVID-19 patient at the town they are living (AOR = 1.81; 95% CI: 1.01,3.28), and not having any reading materials about their profession (AOR = 2.17; 95% CI: 1.15, 4.07) were significantly associated with stress (**Table 5**).

## Discussion

This study aimed at assessing depression, anxiety and stress as well as their associated factors among Ethiopian University students during the early stage of the COVID-19 pandemic. About 46.3% with 95% CI (41.6%, 50.8%) of students reported depression while 52% with 95% CI (47.1%, 56.7%) reported anxiety and about 28.6% with 95% CI (24.6%, 32.9%) reported stress in the current study. Being female and lack of access to reading materials regarding COVID-19 were common risk factors for depression, anxiety and stress. Besides, students who reported to be not well prepared to protect themselves from the pandemic (those with lower self-efficacy), those who have no reading materials at hand about their profession, and those who had no sufficient uninterrupted internet access were more depressed. Students with lower age, those who were from non-health-related fields, and those who do not think that COVID-19 is preventable, and those who had no reading materials at hand were more anxious

whereas study subjects who had no sufficient uninterrupted internet access, those at 1[st] and 2[nd] year and those living in areas where there is confirmed case of COVID-19 were more stressed in the current study.

The prevalence of depression in the current study was lower than reports from Bangladesh [46], Jordan [47] and Pakistan [12]. However, it was higher than studies conducted among university students during the COVID-19 pandemic from Guangzhou, China [48], Spain [18], Iran [49] and European College students [10]. The current prevalence was also higher than studies in Ethiopia among University students before the pandemic [50].

The current prevalence of anxiety was higher than reports from China [48], Jordan [47] and Iran [49]. However, it was lower than studies from Poland [51] and Pakistan [12].

The prevalence of stress in this study was lower than a report from Poland [51] and Pakistan [12].

The variation in the prevalence of depression, anxiety and stress might be attributed to the differences in the socioeconomic conditions, times of the study, the different impact of COVID-19 and the tool used to assess these mental health outcomes. For example in the study from Bangladesh [46] the tool used to assess depression and anxiety were the Patient Health Questionnaire (PHQ-9) and Generalized Anxiety Disorder (GAD-7), respectively. The study from China [48] used the Center for Epidemiologic Studies Depression Scale (CES-D) and Self-Rating Anxiety Scale (SAS) for assessing depression and anxiety, respectively. The tool used to assess stress in Poland [51] was the Perceived Stress Scale (PSS-10). Whereas in Iran [49] authors used Beck's Depression Inventory (BDI-II) and Beck's Anxiety Inventory (BAI) for assessing depression and anxiety, respectively. However, we have used DASS-21 for the assessment of depression, anxiety and stress in the current study.

Female students were more depressed, anxious and stressed in the current study. This finding is consistent with several earlier studies [1, 23, 29, 33–42]. Some evidence [52, 53] suggested that the female reproductive cycle may have a role in the pronounced prevalence of mental illnesses among females. The intensive fluctuations in estrogen and progesterone during the menstrual cycle is related to changes in the hormone's neuroprotective effects, which might escalate the chronicity correlated with mental health problems [52]. This might also be related to the lower risk of developing mental illnesses in males due to differential access to appropriate health care services [54]. Metacognitive beliefs in uncontrollability, advantages and avoidance of worry may also contribute to the higher prevalence of mental illness among females as compared to males [55]. So far, several environmental, genetic and physiological factors were suggested that may play a significant role in the gender differences of mental illness problems [56–58]. However, a study in China revealed the absence of variation in mental health problems based on gender [59].

Students with a lack of access to reading material regarding COVID-19 had higher odds of depression, anxiety and stress. This is clear as more informed study subjects will have reduced levels of mental illnesses as they have sufficient choice to protect themselves from the pandemic [60]. Students with clear information about the cause, route of transmission and prevention mechanisms of the pandemic are less likely to be depressed, anxious and stressed. Those without clear information are liable to misinformation and there is a growing body of literature that reveals the link between misinformation and mental illness [46, 61–64]. Timely and accurate information is fundamental for mitigating and preventing the pandemic [65, 66].

Likewise, study subjects who had no sufficient uninterrupted internet access were more stressed. Those study subjects from the non-health area of study were more anxious in this survey. This is in line with previous studies [16–18]. Students in the health and health-related field are expected to have more appropriate and accurate information and may help them to

get prepared easily to protect themselves from the pandemic which in turn will help to reduce anxiety due to the adverse condition.

Students with lower self-efficacy were more depressed in the current study. Self–efficacy shows to what extent the students are well prepared to cope up with the pandemic. In line with our result, earlier studies also showed that study participants with lower self-efficacy were prone to poor mental health conditions [19–21]. Better self-efficacy has a significant role in promoting desirable performance in the face of adversity conditions [67].

Students with younger ages were more likely to be anxious in this study. This is supported by numerous previous studies [23–32]. The possible explanation might be due to developmental challenges during adolescence that may provoke more anxiety among younger aged adults [68, 69].

Study subjects at 1st and 2nd year and those living in areas where there is a confirmed case of COVID-19 were also more stressed in the current study. This finding is in line with a study done among Spanish students [18]. However, this finding was against a previous result reported among under-graduate students in New Jersey [70].

Finally, this study was the first study in Ethiopia to assess depression, anxiety, and stress among university students at the advent of the COVID-19 pandemic. As a strength, we used a robust and validated tool whereas the result of this survey should be utilized bearing in mind several limitations. The limitations include; lack of generalizability due to a small sample size and absence of random sampling as this study was based on voluntary participation. Besides, social desirability bias and the inherent weakness of the cross-sectional design may result in over or under-report of the symptoms. Since this was a cross-sectional survey whether the heightened level of mental illnesses is due to COVID-19 or other factors is poorly understood.

## Conclusions

A significant proportion of students were affected by common mental illnesses (i.e. depression, anxiety and stress). The level of depression, anxiety, and stress among students are higher as compared to several reports before the pandemic. Numerous factors such as age, gender, self-efficacy, year and field of study, availability of information source and reading materials were identified as factors contributing to either of the common mental health problems. The mental health situation may be improved by the provision of adequate and accurate information and raising the self-efficacy of students.

## Supporting information

**S1 Data. Original data set for depression.**
(SAV)

**S2 Data. Original data set for anxiety.**
(SAV)

**S3 Data. Original data set for stress.**
(SAV)

## Acknowledgments

The authors are grateful for the study participants.

## Author Contributions

**Conceptualization:** Wudneh Simegn, Yigizie Yeshaw, Birhanemeskel Woldegerima, Henok Dagne.

**Data curation:** Henok Dagne.

**Formal analysis:** Wudneh Simegn, Baye Dagnew, Yigizie Yeshaw, Birhanemeskel Woldegerima, Henok Dagne.

**Funding acquisition:** Henok Dagne.

**Investigation:** Wudneh Simegn, Baye Dagnew, Yigizie Yeshaw, Henok Dagne.

**Methodology:** Wudneh Simegn, Baye Dagnew, Yigizie Yeshaw, Sewbesew Yitayih, Birhanemeskel Woldegerima, Henok Dagne.

**Project administration:** Wudneh Simegn, Henok Dagne.

**Resources:** Baye Dagnew, Yigizie Yeshaw, Henok Dagne.

**Software:** Wudneh Simegn, Baye Dagnew, Yigizie Yeshaw, Birhanemeskel Woldegerima, Henok Dagne.

**Supervision:** Baye Dagnew, Yigizie Yeshaw, Henok Dagne.

**Validation:** Yigizie Yeshaw, Sewbesew Yitayih, Birhanemeskel Woldegerima, Henok Dagne.

**Visualization:** Henok Dagne.

**Writing – original draft:** Wudneh Simegn, Baye Dagnew, Yigizie Yeshaw, Sewbesew Yitayih, Birhanemeskel Woldegerima, Henok Dagne.

**Writing – review & editing:** Wudneh Simegn, Baye Dagnew, Yigizie Yeshaw, Sewbesew Yitayih, Birhanemeskel Woldegerima, Henok Dagne.

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
