## [Decision Letter · Decision Letter 0]

12 Jan 2021

PONE-D-20-39731

Depression, Anxiety, Stress and Their Associated Factors among Ethiopian University Students during early stage of COVID-19 pandemic: An online Based Cross-sectional Survey

PLOS ONE

Dear Dr. Dagne,

Thank you for submitting your manuscript to PLOS ONE. After careful consideration, we feel that it has merit but does not fully meet PLOS ONE’s publication criteria as it currently stands. Therefore, we invite you to submit a revised version of the manuscript that addresses the points raised during the review process.

An expert in the field handled your manuscript, and we are appreciative of their time and contributions. Interest was found in your study. The reviewer offers suggestions for improvement. Please address ALL of the reviewer's comments in your revised manuscript and outline in a response-to-reviewer file. 

We look forward to receiving your revised manuscript.

Kind regards,

Frank T. Spradley

Academic Editor

PLOS ONE

2. Please clarify in the Methods section the criterion for participants to be social media users, and why this was in place.

Reviewers' comments:

Reviewer's Responses to Questions

**Comments to the Author**

1. Is the manuscript technically sound, and do the data support the conclusions?

Reviewer #1: Yes

2. Has the statistical analysis been performed appropriately and rigorously? 

Reviewer #1: Yes

3. Have the authors made all data underlying the findings in their manuscript fully available?

Reviewer #1: Yes

4. Is the manuscript presented in an intelligible fashion and written in standard English?

Reviewer #1: Yes

5. Review Comments to the Author

Reviewer #1: I read with great interest this study by Simegn and colleagues on the mental health issues of African students in this on-going pandemic. The topic is no doubt timely, and sheds light on how African youth has coped up in these times. Aside from minor language correction, I suggest including a similar study in the Middle East, whose findings echo this study:

Alfawaz HA, Wani K, Aljumah AA, Aldisi D, Ansari MGA, Yakout SM, Sabico S, Al-Daghri NM. Psychological well-being during COVID-19 lockdown: Insights from a Saudi State University's Academic Community. J King Saud Univ Sci. 2021 Jan;33(1):101262.

I would also recommend the use of a flowchart to give a better picture of the respondents in the survey.

6. PLOS authors have the option to publish the peer review history of their article (what does this mean?). If published, this will include your full peer review and any attached files.

Reviewer #1: No

---

## [Author Response · Author response to Decision Letter 0]

4 Mar 2021

Point by point response for editor/reviewers comments

Manuscript title: Depression, anxiety, stress and their associated factors among Ethiopian University students during early stage of COVID-19 pandemic: An online based cross-sectional survey

Manuscript ID: PONE-D-20-39731

Dear editor/reviewers: Thank you for giving us the chance to revise the manuscript. 

We have addressed all of the concerns raised. These modifications are also incorporated in the revised manuscript.

A. Response to editor comments 

Response: Thank you. We have checked the manuscript for the journal requirements and made all necessary amendments in the revised manuscript. 

2. Please clarify in the Methods section the criterion for participants to be social media users, and why this was in place.

Response: We prefer to use social media users because it enables us to collect the data without direct contact with the study participants, which is important to reduce the rate of spread of the COVID 19 pandemic. Based on your suggestion, we have also included this statement in method section of the revised manuscript (line 109 and 110). 

3. Thank you for stating the following financial disclosure: "The funders had no role in study design, data collection and analysis, decision to publish, or preparation of the manuscript." At this time, please address the following queries: 

a. Please clarify the sources of funding (financial or material support) for your study. List the grants or organizations that supported your study, including funding received from your institution.

d. If you did not receive any funding for this study, please state: “The authors received no specific funding for this work.”

Response: Thank you. The authors received no specific funding for this work. We have also included this statement (see line 319).

4. In your Data Availability statement, you have not specified where the minimal data set underlying the results described in your manuscript can be found

Response: Thank you. We have modified the data availability statement based on your suggestion. 

B. Response to comments of reviewer 1 

1. Aside from minor language correction, I suggest including a similar study in the Middle East, whose findings echo this study:Alfawaz HA, Wani K, Aljumah AA, Aldisi D, Ansari MGA, Yakout SM, Sabico S, Al-Daghri NM. Psychological well-being during COVID-19 lockdown: Insights from a Saudi State University's Academic Community. J King Saud Univ Sci. 2021 Jan;33(1):101262.

Response: we have tried to revise and correct language errors throughout the paper. We have cited the paper you suggested in the revised paper as well (line 74 and 75).

2. I would also recommend the use of a flowchart to give a better picture of the respondents in the survey.

Response: Thank you very much for this suggestion. However, we believe that adding flow chart will increase the volume as we have used five tables and the use of flow chart will not add to much information as we believe.

---

## [Decision Letter · Decision Letter 1]

8 Mar 2021

PONE-D-20-39731R1

Depression, Anxiety, Stress and Their Associated Factors among Ethiopian University Students during early stage of COVID-19 pandemic: An online Based Cross-sectional Survey

PLOS ONE

Dear Dr. Dagne,

Thank you for submitting your manuscript to PLOS ONE. After careful consideration, we feel that it has merit but does not fully meet PLOS ONE’s publication criteria as it currently stands. Therefore, we invite you to submit a revised version of the manuscript that addresses the points raised during the review process.

You need to include a flowchart of participants.

We look forward to receiving your revised manuscript.

Kind regards,

Frank T. Spradley

Academic Editor

PLOS ONE

Journal Requirements:

1) We suggest you thoroughly copyedit your manuscript for language usage, spelling, and grammar. If you do not know anyone who can help you do this, you may wish to consider employing a professional scientific editing service.  

Reviewers' comments:

Reviewer's Responses to Questions

**Comments to the Author**

1. If the authors have adequately addressed your comments raised in a previous round of review and you feel that this manuscript is now acceptable for publication, you may indicate that here to bypass the “Comments to the Author” section, enter your conflict of interest statement in the “Confidential to Editor” section, and submit your "Accept" recommendation.

Reviewer #1: All comments have been addressed

2. Is the manuscript technically sound, and do the data support the conclusions?

Reviewer #1: Yes

3. Has the statistical analysis been performed appropriately and rigorously? 

Reviewer #1: Yes

4. Have the authors made all data underlying the findings in their manuscript fully available?

Reviewer #1: Yes

5. Is the manuscript presented in an intelligible fashion and written in standard English?

Reviewer #1: Yes

6. Review Comments to the Author

Reviewer #1: While the authors have satisfactorily addressed the comments, I still believe a flowchart of participants would have enhanced the paper, given its lack of figures. As it is however it is still acceptable.

7. PLOS authors have the option to publish the peer review history of their article (what does this mean?). If published, this will include your full peer review and any attached files.

Reviewer #1: No

---

## [Author Response · Author response to Decision Letter 1]

26 Apr 2021

Response to editor’s comments (PONE-D-20-39731R2)

Dear editor, thank you very much for your comments regarding our manuscript. 

Based on your request we have included the study participants’ flowchart and revised the manuscript for the language related issues thoroughly. 

We have enclosed herewith the manuscript clean and track change versions along with the flow chart Tif format.

Thank you!

---

## [Editor Report · Decision Letter 2]

30 Apr 2021

Depression, anxiety, stress and their associated factors among Ethiopian University students during an early stage of COVID-19 pandemic:  An online-based cross-sectional survey

PONE-D-20-39731R2

Dear Dr. Dagne,

We’re pleased to inform you that your manuscript has been judged scientifically suitable for publication and will be formally accepted for publication once it meets all outstanding technical requirements.

Kind regards,

Frank T. Spradley

Academic Editor

PLOS ONE

---

## [Editor Report · Acceptance letter]

21 May 2021

PONE-D-20-39731R2 

Depression, anxiety, stress and their associated factors among Ethiopian University students during an early stage of COVID-19 pandemic:  An online-based cross-sectional survey 

Dear Dr. Dagne:

I'm pleased to inform you that your manuscript has been deemed suitable for publication in PLOS ONE. Congratulations! Your manuscript is now with our production department. 

Kind regards, 

on behalf of

Dr. Frank T. Spradley 

Academic Editor

PLOS ONE